# The Mutational Landscape of Acute Myeloid Leukaemia Predicts Responses and Outcomes in Elderly Patients from the PETHEMA-FLUGAZA Phase 3 Clinical Trial

**DOI:** 10.3390/cancers13102458

**Published:** 2021-05-18

**Authors:** Rosa Ayala, Inmaculada Rapado, Esther Onecha, David Martínez-Cuadrón, Gonzalo Carreño-Tarragona, Juan Miguel Bergua, Susana Vives, Jesus Lorenzo Algarra, Mar Tormo, Pilar Martinez, Josefina Serrano, Pilar Herrera, Fernando Ramos, Olga Salamero, Esperanza Lavilla, Cristina Gil, Jose Luis López Lorenzo, María Belén Vidriales, Jorge Labrador, José Francisco Falantes, María José Sayas, Bruno Paiva, Eva Barragán, Felipe Prosper, Miguel Ángel Sanz, Joaquín Martínez-López, Pau Montesinos

**Affiliations:** 1Hematology Department, Hospital Universitario 12 de Octubre, Instituto de Investigación Sanitaria Imas12, 28041 Madrid, Spain; inmaculada.rapado@salud.madrid.org (I.R.); mariaesther.onecha@scsalud.es (E.O.); gonzalo.carreno@salud.madrid.org (G.C.-T.); 2Hematological Malignancies Clinical Research Unit, CNIO, 28029 Madrid, Spain; 3Departament of Medicine, Complutense University, 28040 Madrid, Spain; 4Centro de Investigación Biomédica en Red de Cáncer (CIBERONC), Instituto Carlos III, 28029 Madrid, Spain; bpaiva@unav.es (B.P.); barragan_eva@gva.es (E.B.); fprosper@unav.es (F.P.); Miguel.Sanz@uv.es (M.Á.S.); montesinos_pau@gva.es (P.M.); 5Hematology Department, Hospital Universitario y Politécnico La Fe, 46026 Valencia, Spain; martinez_davcua@gva.es; 6Hematology Department, Hospital San Pedro Acantara, 10003 Cáceres, Spain; juanmiguel.bergua@salud-juntaex.es; 7Department of Hematology, ICO Badalona-Hospital Germans Trias i Pujol. Josep Carreras Leukemia Research Institute. Universitat Autònoma de Barcelona, 08916 Badalona, Spain; svives@iconcologia.net; 8Hematology Department, Hospital General de Albacete, 02006 Albacete, Spain; jlalgarra@sescam.jccm.es; 9Hematology Department, Hospital Clínico Universitario de Valencia, 46010 Valencia, Spain; tormo_mar@gva.es; 10Hematology Department, Hospital 12 de Octubre, 28041 Madrid, Spain; mariapilar.martinez.sanchez@salud.madrid.org; 11Hematology Department, Hospital Universitario Reina Sofía, 14004 Cordoba, Spain; josefina.serrano.sspa@juntadeandalucia.es; 12Hematology Department, Hospital Ramon y Cajal, 28034 Madrid, Spain; pilar.herrera@salud.madrid.org; 13Hematology Department, Hospital Universitario de León, 24008 León, Spain; mail@fernandoramosmd.es; 14Hematology Department, Hospital Universitari Vall d’Hebron, 08035 Barcelona, Spain; osalamer@vhebron.net; 15Hematology Department, Hospital Universitario Xeral de Lugo, 27003 Lugo, Spain; esperanza.Lavilla.Rubira@sergas.es; 16Hematology Department, Hospital General de Alicante, 03010 Alicante, Spain; gil_cricor@gva.es; 17Hematology Department, Fundación Jiménez Díaz, 28040 Madrid, Spain; jllopez@quironsalud.es; 18Hematology Department, Hospital Universitario de Salamanca, IBSAL, 37007 Salamanca, Spain; mbvidriales@saludcastillayleon.es; 19Hematology Department, Hospital Universitario de Burgos, 09001 Burgos, Spain; jlabradorg@saludcastillayleon.es; 20Hematology Department, Hospital Universitario Vírgen del Rocío, Instituto de BioMedicina de Sevilla, 41013 Sevilla, Spain; josef.falantes.sspa@juntadeandalucia.es; 21Hematology Department, Hospital Doctor Peset, 46017 Valencia, Spain; sayas_mjo@gva.es; 22Hematology Department, Clínica Universitaria de Navarra, 31008 Navarra, Spain

**Keywords:** clinical trials and observations, myeloid neoplasia, NGS, variants, leukemia, myelocytic, acute, leukemic cells, older adults, genetic risk, complete remission, cytarabine, azacytidine, prognostic factors

## Abstract

**Simple Summary:**

Mutational profiling using a custom 43-gene next-generation sequencing panel revealed that patients with mutated *DNMT3A* or *EZH2*, or an increase in *TET2* VAF and lower *TP53* VAF showed a higher overall response. *NRAS* and *TP53* variants were associated with shorter overall survival (OS), whereas only mutated *BCOR* was associated with a shorter relapse-free survival (RFS). Subgroup analyses of OS according to biological and genomic characteristics showed that patients with low–intermediate cytogenetic risk and mutated *NRAS* benefited from azacytidine therapy and patients with mutated *TP53* showed a better RFS in the azacytidine arm. In conclusion, differential mutational profiling might anticipate the outcomes of first-line treatment choices (AZA or FLUGA) in older patients with AML.

**Abstract:**

We sought to predict treatment responses and outcomes in older patients with newly diagnosed acute myeloid leukemia (AML) from our FLUGAZA phase III clinical trial (PETHEMA group) based on mutational status, comparing azacytidine (AZA) with fludarabine plus low-dose cytarabine (FLUGA). Mutational profiling using a custom 43-gene next-generation sequencing panel revealed differences in profiles between older and younger patients, and several prognostic markers that were useful in young patients were ineffective in older patients. We examined the associations between variables and overall responses at the end of the third cycle. Patients with mutated *DNMT3A* or *EZH2* were shown to benefit from azacytidine in the treatment-adjusted subgroup analysis. An analysis of the associations with tumor burden using variant allele frequency (VAF) quantification showed that a higher overall response was associated with an increase in *TET2* VAF (odds ratio (OR), 1.014; *p* = 0.030) and lower *TP53* VAF (OR, 0.981; *p* = 0.003). In the treatment-adjusted multivariate survival analyses, only the *NRAS* (hazard ratio (HR), 1.9, *p* = 0.005) and *TP53* (HR, 2.6, *p* = 9.8 × 10^−7^) variants were associated with shorter overall survival (OS), whereas only mutated *BCOR* (HR, 3.6, *p* = 0.0003) was associated with a shorter relapse-free survival (RFS). Subgroup analyses of OS according to biological and genomic characteristics showed that patients with low–intermediate cytogenetic risk (HR, 1.51, *p* = 0.045) and mutated *NRAS* (HR, 3.66, *p* = 0.047) benefited from azacytidine therapy. In the subgroup analyses, patients with mutated *TP53* (HR, 4.71, *p* = 0.009) showed a better RFS in the azacytidine arm. In conclusion, differential mutational profiling might anticipate the outcomes of first-line treatment choices (AZA or FLUGA) in older patients with AML. The study is registered at ClinicalTrials.gov as NCT02319135.

## 1. Introduction

Older patients with acute myeloid leukemia (AML), defined as those aged 65 or beyond, who are unsuitable for standard induction therapy generally has a poor prognosis, and fewer than 10% survive beyond 3 years. This is due to the often unfavorable genetic profiles of elderly patients, the presence of comorbidities that aggravate the side-effects of therapy, and the very high relapse rates, even for those who achieve complete morphological remission (CR) [1]. For elderly patients treated with intensive chemotherapy, the overall survival (OS) at 1, 2, and 3 years is 30%, 15%, and <5%, respectively [2].

Recent studies in elderly patients have tested the effects of low-intensity chemotherapy using hypomethylating agents (HMAs) such as azacytidine and decitabine, which are less toxic than conventional chemotherapeutic schemes. The results showed a CR rate of 15–20% and an improved OS when compared with supportive treatment or low-dose cytarabine (LDAC) [3,4]. The survival benefit of low-intensity HMA therapy is not limited to patients with morphological CR [5].

The clinical outlook for the elderly population has improved over the last few years with the approval of venetoclax and other targeted therapies, such as *FLT3* inhibitors (midostaurin and gilteritinib, among others), IDH1/2 inhibitors (ivosidenib and enasidenib, respectively), and other novel therapies such as hedgehog pathway inhibitors (e.g., glasdegib) [6]. This change in the treatment paradigm makes it imperative to develop new biomarkers that help guide treatment approaches and improve the outcome in elderly patients with AML.

While knowledge of the molecular landscape in AML continues to evolve, the majority of studies are retrospective in nature. Consequently, treatment strategies are heterogeneous, and the clinical applicability of this new molecular information is limited. It is, therefore, essential to perform well-designed clinical trials that include associated molecular studies in order to test new biomarkers of responses.

*TET2* mutations predict responses to HMAs in myelodysplastic syndrome (MDS) [7]; however, this has not been substantiated in AML. *TP53* mutations have been previously associated with better responses to HMAs in AML [8], but there are no molecular biomarkers for LDAC-based or fludarabine schemes.

Our aim was to predict the responses and outcomes in older patients with AML at the time of diagnosis via mutation status in the context of the FLUGAZA phase III clinical trial (NCT02319135), which compares 5′-azacytidine (AZA arm) with LDAC plus fludarabine (FLUGA arm) treatment in newly diagnosed AML patients.

## 2. Patients and Methods

### 2.1. Identification Cohort

We analyzed bone marrow (BM) samples at the time of diagnosis from 207 of 285 patients with AML treated in accordance with the FLUGAZA trial (Appendix A) (AZA arm (*n* = 96) and FLUGA arm (*n* = 111)). DNA at diagnosis was not available for 78 cases. The patients included in the AZA arm received 3 induction cycles of azacytidine followed by 6 consolidation cycles. The patients included in the FLUGA arm were randomized to receive 3 induction cycles of cytarabine plus fludarabine (FLUGA), followed by 6 cycles of reduced-intensity fludarabine and LDAC. Treatment was continued for 9 more cycles, unless the minimal residual disease as assessed by flow cytometry was negative. The median age at diagnosis was 75 years (range, 65–90) in the sub-study cohort. Both treatment groups were balanced for age, leukocyte count, baseline BM blasts, karyotype risk (ELN-2017 [9]), and, also, for *FLT3*-internal tandem duplication (*FLT3*-ITD) and mutated *NPM1*. The main clinical characteristics of the patients included in the sub-study are summarized in Table 1 and Appendix A. The baseline characteristics and efficacy outcomes in the subset of 207 patients in whom molecular assessments were performed were similar to those of the 78 excluded patients who did not undergo molecular assessment (Appendix A).

All patients provided written informed consent, and the trial was approved by the appropriate institutional review boards or ethics committees of the participating institutions. The study was registered at www.ClinicalTrials.gov as NCT02319135 (accessed on 6 April 2020), and the results of the FLUGAZA trial have been published elsewhere [10,11].

### 2.2. Methods

#### 2.2.1. High-Sensitivity Targeted Sequencing and Mutation Analysis

DNA was extracted using a Maxwell^®^ 16 MDx instrument (Promega Biotech Iberica SL, Madrid, Spain) and quantified on a Qubit^®^ 2.0 Fluorometer (Invitrogen, Thermo Fisher Scientific Inc., Waltham, MA, USA). Library preparation was carried out according to the manufacturer’s protocol (Life Technologies, Palo Alto, CA, USA). High-depth next-generation sequencing (NGS) was performed on an Ion Torrent S5^XL^ sequencer (Life Technologies).

Mutational profiling was performed via targeted NGS using a custom panel of 43 genes implicated in myeloid pathology (described in the Appendix A). The total number of reads obtained in each sample was two million, with an average depth of coverage >2000 reads per nucleotide and high uniformity amongst all fragments (92%). Data analyses were performed using Ion Reporter v4.4 software (Life Technologies, Carlsbad, CA, USA), which identified single nucleotide variants (SNV) and small insertions or deletions (InDels). We employed Ion Reporter default parameters and filtered out variants with a total coverage of at least 70 reads and a variant allelic coverage of at least 10 reads. Variants with a minor allelic frequency >0.01 in the general population according to the single nucleotide polymorphism database (NCBI, dbSNP150) and/or the 5000-exome sequencing project were also rejected as possible polymorphisms (https://evs.gs.washington.edu/EVS; accessed on 20 December 2019). Filtered variants were then annotated using the Catalogue of Somatic Mutations in Cancer (COSMIC) database (https://cancer.sanger.ac.uk/census; accessed on 20 December 2019), allowing those variants present in some tumors to be retained, and those present in dbSNP and previously identified as cancer mutations to be retained. Filtered variants that were absent from dbSNP or COSMIC but were “deleterious” due to associated functional changes at the protein level, or due to their occurrence in conserved regions, were considered in the final analysis. All the mutations included in this study are listed in Appendix A.

#### 2.2.2. Availability of Data and Materials

All supporting data are included in the manuscript and Appendix A. The data discussed in this publication have been deposited in the NCBI Sequence Read Archive (SRA) and are accessible via the following link: https://www.ncbi.nlm.nih.gov/sra/PRJNA655113 (accessed on 6 August 2020). Additional data are available upon reasonable request from the corresponding author.

#### 2.2.3. Statistical Analysis

Statistical analyses were performed with SPSS v22.0 software (SPSS Inc., Chicago, IL, USA). The clinical characteristics of the patients were compared using the Chi-squared test for categorical variables and Student’s *t*-test for continuous variables. The relationship between clinical variables, including mutational status, and response to azacytidine or FLUGA was evaluated using multivariate logistic regression analysis. Relapse-free survival (RFS) was defined as the time from the first CR after diagnosis to relapse, death, or the date of the last follow-up. OS was calculated from the date of AML diagnosis to death or the last follow-up date. Cox proportional hazard models and Kaplan–Meier analyses were used to assess the association of variables (clinical data and mutational profile) with patient RFS and OS, and were adjusted via treatment received. For multivariate analyses, patient age (continuous variable), leukocytes (categorized variable), cytogenetic risk (low–intermediate vs. high risk), ELN-2017 classification [9], number of mutations (0–8), mutated genes in panel (yes/no), and DTA (*DNMT3A*-*TET2*-*ASXL1*) mutations were included. Multiple comparisons were adjusted by Bonferroni correction. We used overall treatment response, which includes partial remission (PR) and CR/CRi (complete remission with incomplete recovery of blood counts), to define subgroups that are clinically relevant in the investigation of molecular markers associated with treatment sensitivity and resistance. In all cases, statistical significance was considered at a *p*-value less than or equal to 0.05. 

## 3. Results

The mutational profile of older patients with AML is different to that of younger patients, and several prognostic markers have no impact in older patients. We detected a total of 893 variants—247 small InDels and 646 SNV. In total, 98% of patients (*n* = 203) presented at least one detectable mutation, with a median number of mutations of four (range: zero to eight). The most commonly mutated genes were *TET2* (*n* = 55), *FLT3* (*n* = 52), SRSF2 (*n* = 49), *TP53* (*n* = 45), *DNMT3A* (*n* = 45), *ASXL1* (*n* = 45), *RUNX1* (*n* = 43), *IDH2* (*n* = 36), *IDH1* (*n* = 34), *NPM1*, (*n* = 33), and *NRAS* (*n* = 23).

Remarkably, the mutational profiles of older patients differed considerably from those of a previous series of young patients with AML treated almost entirely with a standard high-dose cytarabine regimen plus idarubicin (3 + 7 regimen) (Appendix A) [12,13,14].

We evaluated conventional molecular markers with an impact on OS, but no marker with prognostic impact was detected in our cohort (Appendix A). Indeed, recurrent genetic abnormalities (*FLT3*-ITD, *NPM1,* and *CEBPA* mutations) had no significant impact on OS as assessed via Cox multivariate analyses adjusted by treatment, unlike what was seen in younger AML patients. Patients with low–intermediate risk clearly benefitted from AZA (median overall survival 14 months in the AZA arm vs. 6 months in the FLUGA arm; *p* = 0.003) (Appendix A). Only cytogenetic risk had a prognostic impact on OS (hazard ratio (HR) 1.67; 95% confidence interval (CI) 1.15–2.43; *p* = 0.007) as assessed via COX multivariate analyses, irrespective of treatment received, but it had no impact on RFS.

### 3.1. Mutational Landscape Predicts Response to Azacytidine and LDAC Plus Fludarabine Treatments

The mutational landscapes in responders (patients who achieved CR or CRi) and non-responders under each of the two treatments are shown in Appendix A. CR/CRi was achieved after the third cycle in 24% of patients in the AZA arm, and in 28% of patients in the FLUGA arm. Treatment-adjusted logistic regression analysis identified the following predictive markers that were associated with achieving CR after the third cycle of azacytidine or FLUGA (Appendix A): lower patient age (odds ratio (OR), 0.92; *p* = 0.037), mutated *KMT2A* (OR, 6.68; *p* = 0.006), mutated *NF1* (OR, 9.01; *p* = 0.005), mutated *PHF6* (OR, 7.93; *p* = 0.014), mutated *U2AF1* (OR, 7.09; *p* = 0.04), wild-type *NRAS* (OR, 0.04; *p* = 0.021), and wild-type *TP53* (OR, 0.18; *p* = 0.039).

The main response associated with a longer OS in this series of AML patients was overall response, which includes PR and CR/CRi (*p* < 0.001, Appendix A). The overall response rate (ORR) was 40% after the third cycle of azacytidine and 37% after the third cycle of FLUGA. Treatment-adjusted logistic regression analysis identified the following predictive markers associated with ORR after the third cycle of azacytidine or FLUGA (Appendix A): lower patient age (OR 0.94; *p* = 0.078), wild-type *TP53* (OR 0.19; *p* = 0.018), mutated *KMT2A* (OR 4.29; *p* = 0.025), mutated *NF1* (OR 4.53; *p* = 0.033), and mutated *TET2* (OR 2.35; *p* = 0.07). Neither azacytidine nor FLUGA therapy impacted the ORR. Moreover, multivariate analyses showed that no variable was independently associated with achieving overall response after Bonferroni adjustment for multiple comparisons. Nevertheless, when we performed a subgroup analysis of treatment responders (defined as patients achieving overall response) via biological and genomic characteristics (Figure 1), we observed that patients with mutated *DNM3A* or *EZH2* benefited from azacytidine.

We defined a molecular signature that predicts response to azacytidine based on the presence of mutations in *EZH2, U2AF1, DNMT3A,* or *TET2* (Figure 1). This signature was selected based on our previous data and descriptions in the literature (e.g., *TET2*). This signature was detected in 106 cases, of which 25/48 patients in the AZA arm (52%) and 19/58 (33%) in the FLUGA arm achieved OR (*p* = 0.044). The signature predicted OR after the third azacytidine cycle with a sensitivity of 64% and a specificity of 60%, and positive and negative likelihood ratios of 1.6 and 0.6, respectively. Likewise, the signature predicted CR after the third azacytidine cycle with a sensitivity of 78% and specificity of 59%, and positive and negative likelihood ratios of 1.9 and 0.4, respectively.

### 3.2. The Higher Variant Allele Frequency of Some Variants Influences Overall Response after the Third Cycle of Azacytidine and LDAC Plus Fludarabine

In our cohort, the median variant allele frequency (VAF) was higher in the variants detected in *ASXL1, DNMT3A, RUNX1, SRSF2, TET2,* and *TP53* genes, while genes implicated in signaling pathways (*CALR*, *CBL*, *EPOR*, *KIT*, *KRAS*, *MPL*, *NRAS,* and *THPO*), such as *KDM6A*, *PHF6,* and *SH2B3*, showed lower median VAFs (Appendix A).

Univariate analysis revealed that the VAF distribution was significantly different between responders and non-responders in both treatment arms (AZA arm, VAF-*TET2* (20% vs. 8.7%, *p* = 0.022) and VAF-*TP53* (5.5% vs. 27%, *p* = 0.015); FLUGA arm, VAF-*NRAS* (1.2% vs. 6.7%, *p* = 0.009) and VAF-*TP53* (8.2% vs. 19.6%, *p* = 0.046) (Appendix A). No differences in VAF were observed in the remainder of the genes in the panel.

Treatment-adjusted logistic regression analysis identified the following predictive markers as associated with a higher OR after the third cycle of azacytidine or FLUGA: increases in *TET2* VAF (OR 1.01; 95%CI: 1.001–1.026; *p* = 0.030) and decreases in *TP53* VAF (OR 0.98; 95%CI: 0.969–0.994; *p* = 0.003).

### 3.3. Somatic Mutations in NRAS, TP53, and BCOR Predict a Shorter Overall and/or Relapse-Free Survival According to Univariate Analyses

The median OS in the global series was 15 months (range 1–38). The median OS within treatment arms, in terms of mutant vs. wild-type for those genes mutated in ≥4% of patients, is shown in Appendix A. According to the univariate analyses, *NRAS* and *TP53* mutations were factors adversely affecting OS (Figure 2a,b), and *BCOR* mutations adversely affected RFS (Figure 2c). In the subgroup analysis of OS via biological and genomic characteristics (Figure 3a), we observed that patients with low–intermediate cytogenetic risk and mutated *NRAS* benefited from azacytidine therapy. However, in the subgroup analysis of RFS (Figure 3b), patients with mutated *TP53* displayed a better RFS in the AZA arm. The difference in the mutational profile between the two arms is shown in Appendix A and Appendix A.

We identified the presence of mutations in *NRAS* or *TP53* as a high molecular risk signature, which was detected in 64 patients (31%). In the AZA arm, seven patients (25%) showing high molecular risk achieved an OR after the third cycle, compared with only three patients (8%) in the FLUGA arm (*p* = NS). The high molecular risk condition was associated with unfavorable outcomes and shorter survival times in both arms for OS (Figure 4). The signature predicted OS with a sensitivity of 37% and specificity of 93%, and positive and negative likelihood ratios of 5.29 and 0.68, respectively.

Mutated *NRAS* (HR 1.94, *p* = 0.005) and *TP53* (HR 2.57, *p* < 0.0001) were the only variables associated with a higher risk of death; notably, mutated *BCOR* was the only variable associated with a higher risk of relapse (HR 3.60, *p* = 0.0003) (Table 2). No other variables were identified as predictors of death or leukemia relapse, and no other differences were observed in the Cox multivariate analysis adjusted by treatment. Data of Cox univariate analyses is showed in Appendix A.

## 4. Discussion

Using NGS-based molecular profiling, we have identified a subset of older patients who benefited from azacytidine therapy within the FLUGAZA clinical trial, which compared azacytidine with LDAC plus fludarabin (FLUGA). We have shown that azacytidine is more effective at achieving responses in older patients who present with mutations in *EZH2* or *DNMT3A* at the time of diagnosis. On the other hand, *NRAS* and *TP53* mutations were adverse prognostic factors in the context of OS, and *BCOR* mutations conferred a high risk of leukemia relapse, regardless of treatment received. Patients with low–intermediate cytogenetic risk and mutated *NRAS* benefited from azacytidine therapy for OS, and patients with mutated *TP53* also had a better RFS when receiving azacytidine.

The influence of *NRAS* mutations in response to LDAC has been reported previously; for example, Bloomfield and colleagues showed that patients with AML carrying mutant *RAS* benefit from high-dose cytarabine (HDAC) consolidation more so than patients with wild-type *RAS* [15], and they also exhibit a lower relapse risk (HR 0.28, *p* = 0.002) than patients with mutant *RAS* treated with LDAC. However, *NRAS* mutations also confer resistance to newly targeted drugs, including enasidenib, an IDH2 inhibitor approved for use in refractory/relapsed AML [16]. Indeed, the co-occurrence of *NRAS* and *IDH2* mutations with other MAPK pathway effects was enriched in non-responders, which was consistent with RAS signaling contributing to primary therapeutic resistance. A similar outcome was also seen in clinical trials of *FLT3* inhibitors, wherein *NRAS* mutations were enriched in poor responders to crenolanib or gilteritinib [17,18]. We showed here that *NRAS* mutations confer resistance to LDAC squeme, and patients with these mutations benefited from azacytidine therapy.

We found that *TP53* mutations had no impact on the responses of patients in the context of treatment received, which contradicts the study that described an association between *TP53* mutations and response to 10-day decitabine [8], and which showed higher response rates in mutated *TP53* than wild-type *TP53* patients (100% vs. 41%, *p* < 0.001), although the responses were not long-lasting. However, we derived very few responses from the patients in this trial with mutated *TP53*. Some authors doubt the utility of these predictive biomarkers as improved response rates do not translate to improved survival [19]. However, we also found shorter OS for patients with *NRAS* or *TP53* mutations, and the effects of *NRAS* and *TP53* have previously been described with conventional care regimens [20].

We have defined a molecular signature for identifying responders to azacytidine therapy, characterized by the presence of mutations in *DNMT3A, TET2, EZH2*, or *U2AF1*. In this context, mutations in *TET2* have previously been associated with responses to HMAs in MDS, or in AML with 20–30% blasts [7,21]. Likewise, via a genomics-informed computation biology platform recently developed to predict responses to HMAs in 18 patients with MDS/AML, the authors found that gain-of-function mutations in *EZH2* and *IDH1*/*2* were predictors of the response to the CpG-methylating effects of azacytidine via DNMT1 inhibition [22]. In contrast to our data, Jung and colleagues found that *U2AF1* mutation was significantly associated with non-response to azacytidine in patients with MDS [23]. Our results, however, affirm those of Bejar and colleagues [7], and the response maintained in the ninth cycle was associated with not only the presence of *TET2* mutations, but also the absence of *TP53* mutations.

*BCOR* mutations were associated with reduced RFS in both arms of the FLUGAZA clinical trial. *BCOR* encodes a transcription regulatory factor that controls myeloid proliferation and differentiation. The impacts of *BCOR* or *BCORL1* mutations have previously been described in 28 of the 377 de novo AML cases (7%) [24], and were independent unfavorable prognostic factors in both OS (*p* = 0.004) and RFS (*p* = 0.046).

Our study has some limitations that need to be considered. First, not all the patients included in the trial had samples available for NGS analysis, causing a potential selection bias. Furthermore, *CEBPA* mutations were identified by NGS, and were not classified separately as monoallelic or biallelic mutations. Finally, the presence of mutations in some genes was infrequent (e.g., *FLT3*-ITD and *CEBPA*), and the impact of the co-occurrence of gene mutations was not evaluated due to the low number of cases studied.

Venetoclax plus azacytidine or LDAC is being increasingly used in patients with AML who are >75 years of age and/or not candidates for intensive remission induction therapy. It achieves good CR/CRi rates in high-risk patients, such as those with poor cytogenetics [25], or with *RUNX1* and *ASXL1* mutations [26]. In addition, mutations in *TP53* or *NRAS* also result in shorter OS in relapse/refractory AML treated with venetoclax-based therapy [26]. Based on our results, we would recommend venetoclax plus azacytidine over venetoclax plus LDAC for patients with *TP53* or *NRAS* mutations. We believe that the favorable profile defined here for azacytidine response could also be useful for identifying good responders to an azacytidine plus venetoclax regimen, but this should be confirmed further.

In conclusion, the mutational profile of AML in elderly patients is different from that described in young patients, and the prognostic and predictive markers are also different in this older population. We have defined molecular signatures for identifying patients with poor prognosis associated with azacytidine- or LDAC-based regimens (Table 3). The NGS analysis of targeted gene panels seems useful for identifying prognostic factors and predicting responses in elderly patients with AML.

## Figures and Tables

**Figure 1 cancers-13-02458-f001:**
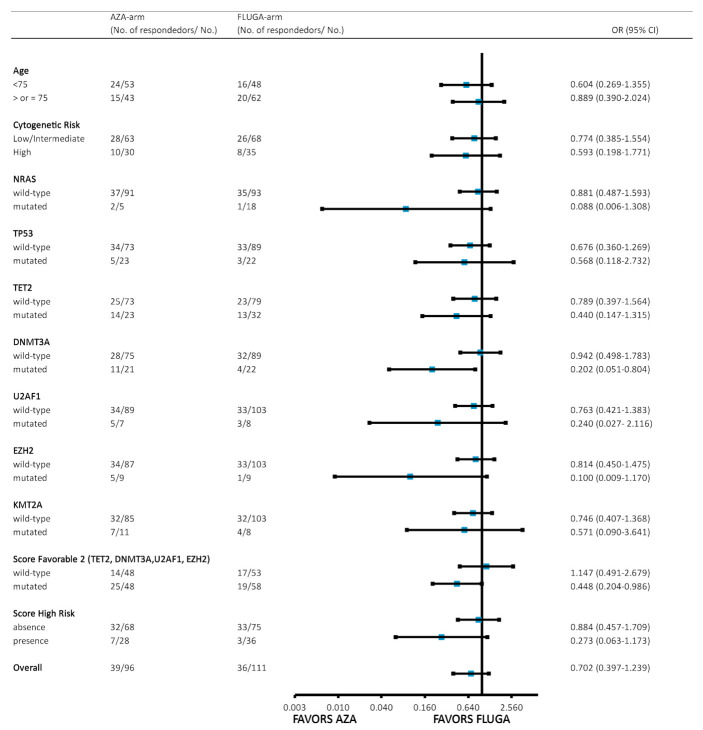
Subgroup analysis of responders to treatment via biological and genomic characteristics. OR: odds ratio, cytogenetic risk: low–intermediate vs. high risk as per ELN 2017 classification; high risk score was defined by the presence of mutated *NRAS* or *TP53*. A score predicting an AZA response was defined by the presence of mutated *EZH2*, *U2AF1*, *DNMT3A,* or *TET2* genes. Patients with baseline mutations in *DNMT3A* (odds ratio (OR) 0.20, *p* = 0.023) or a score predicting AZA response (OR 0.448, *p* = 0.046) could benefit from azacytidine.

**Figure 2 cancers-13-02458-f002:**
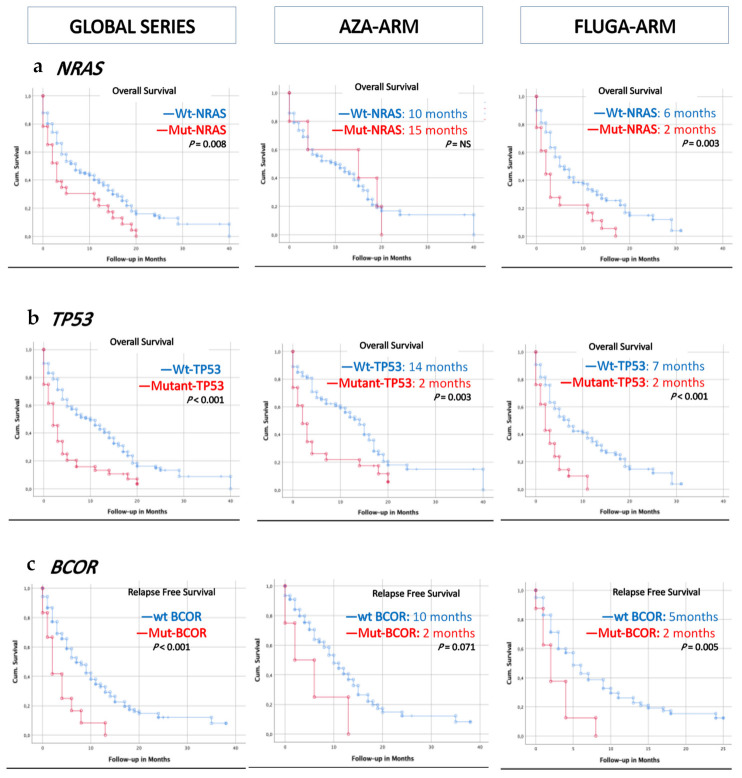
Mutation status has prognostic significance for overall survival and relapse-free survival. In the global series, the AZA arm’s and FLUGA arm’s overall survival and disease free-survival are represented by Kaplan–Meier plots. Wild-type status is indicated in blue and mutated status is indicated in red. *NRAS* mutations (**a**) and *TP53* mutations (**b**) are adverse factors affecting overall survival in the FLUGA arm. *BCOR* mutations are adverse factors affecting relapse-free survival in the AZA and FLUGA arms (**c**). AZA: azacytidine, FLUGA: fludarabine plus low-dose cytarabine (LDAC).

**Figure 3 cancers-13-02458-f003:**
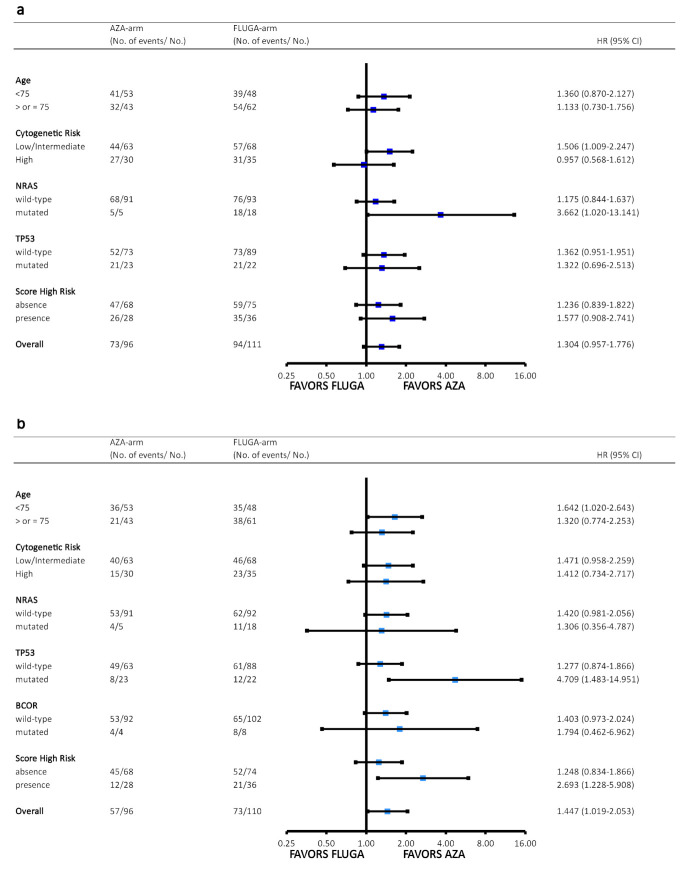
Subgroup analysis of overall survival and progression-free survival via biological and genomic characteristics. HR: hazard ratio, cytogenetic risk: low–intermediate vs. high risk as per ELN 2017 classification; a high-risk score was defined by the presence of mutated *NRAS* or *TP53*. (**a**) in the subgroup analyses of overall survival via biological and genomic characteristics, we observed that patients with low–intermediate cytogenetic risk (hazard ratio (HR0 1.51, *p* = 0.045) and mutated *NRAS* (HR 3.66, *p* = 0.047) could benefit from azacytidine. (**b**) in the subgroup analyses of relapse-free survival, we observed that patients with mutated *TP53* (HR 4.71, *p* = 0.009) and scores indicating a high risk for AML (HR 2.69, *p* = 0.013) showed a higher chance of relapse-free survival under the AZA arm.

**Figure 4 cancers-13-02458-f004:**
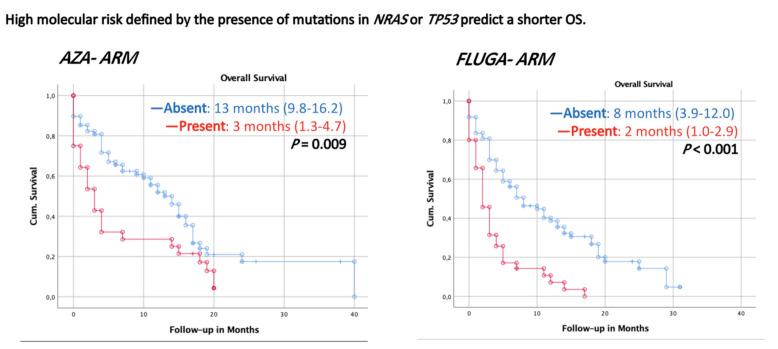
Kaplan–Meier overall survival curves in acute myeloid leukemia (AML) patients classified by the presence or absence of high molecular risk score (HMR). HMR pattern defined by presence of *NRAS* or *TP53* mutations which is associated with unfavorable outcomes and shorter survival after azacytidine (**left**) or FLUGA (**right**) schemes. HMR absent is indicated in blue and HMR present is indicated in red. Number of censored patients with respect to the stratified groups and the number at risk is indicated. *p* values are considered significant (*p* < 0.05). OS: overall survival.

**Table 1 cancers-13-02458-t001:** Patient demographics and baseline characteristics.

Variable		AZA Arm (*N* = 96)	FLUGA Arm (*N* = 111)
Age at diagnosis	Years, median (range)	75 (65–90)	76 (65–88)
Blasts at diagnosis	%, median	55	53
WBC at diagnosis	×10^−9^/L, median	22	21
Dyserythropoiesis	n cases, %	45	47
Dysmyelopoiesis	n cases, %	38	42
Dysthrombopoiesis	n cases, %	23	31
AML origin	de novo	*44*	40
	AML secondary MDS	47	45
	AML secondary Treatment	5	11
FAB classification	M0/M1/M2/M4/M5/M6/M7/NOS	16/15/13/1/21/12/5/9	16/21/22/0/22/12/5/10
Cytogenetics	Abnormal Karyotype/Normal Karyotype	46/38	51/26
Cytogenetics Risk Group	Low–Intermediate Risk	63	68
	High Risk	30	35
WHO classification	AML with certain genetic abnormalities	5	13
	AML with myelodysplastic-related changes	47	45
	AML related to chemotherapy or radiation previous	5	11
	AML NOS	38	42
Follow-up time	Months, median (SD)	15 (9)	16 (7)

AML: acute myeloid leukemia, WBC: white blood cells, AZA: azacytidine, FLUGA: fludarabine plus low-dose cytarabine (LDAC), FAB: French–American–British classification, MO: myeloblastic without cytological maturation, M1: myeloblastic with minimal maturation, M2: myeloblastic with significant maturation, M4: acute myelomonocytic leukemia, M5: acute monoblastic leukemia, M6: acute erythroid leukemia, M7: acute megakaryoblastic leukemia, WHO: World Health Organisation, AML NOS: AML not otherwise specified, cytogenetic risk group: low–intermediate vs. high-risk, classification as per ELN 2017, SD: standard deviation. This study was registered at www.ClinicalTrials.gov as NCT02319135 (accessed on 6 April 2020)

**Table 2 cancers-13-02458-t002:** Biomarkers consistently associated with death or relapse.

Variable	HR	Risk of Death 95% CI for HR		*p*-Value (Bonferroni)
		Lower	Upper	
*NRAS* (wt vs. mut)	1.94	1.21	3.08	0.005 (0.067)
*TP53* (wt vs. mut)	2.57	1.76	3.76	9.8 × 10^−7^ (0.128 × 10^−5^)
**Variable**	**HR**	**Risk of Relapse 95% CI for HR**		***p*-Value (Bonferroni)**
		**Lower**	**Upper**	
*BCOR* (wt vs. mut)	3.60	1.81	7.16	0.000271 (0.004)

Biomarkers identified with an adjusted Cox regression analysis (model; *p* ≤ 0.05) were included in the table. Cox regression model was adjusted for age, cytogenetic risk group (low–intermediate/high risk, classification ELN 2017), AZA or FLUGA arm, and presence of mutations in any gene included in the panel. Analyses based on 194 patients and 157 events for overall survival, and 194 patients and 124 events for relapse-free survival. Wt: wild-type, mut: mutated, HR: hazard ratio, CI: confidence interval; *p*-value is adjusted by the Bonferroni criteria in parenthesis.

**Table 3 cancers-13-02458-t003:** Biomarkers identified in a phase 3 AML trial.

Parameter	Global Series	Favors AZA-Arm *
Predictive markers for response to treatment	Lower patient age Wildtype *TP53* Mutated *KMT2A, NF1* or *TET2*	Mutated *DNMT3A* Presence Score predicting an AZA response
Prognostic markers for OS	Mutated *NRAS* or *TP53* confer adverse prognostic for OS	Mutated *NRAS* Low-Intermediate Cytogenetic Risk.
Prognostic markers for RFS	Mutated *BCOR* confers adverse prognostic for RFS	Mutated *TP53* Presence high molecular risk (HMR)

Summary table of biomarkers identified in a phase 3 AML trial of azacitidine (AZA) vs. low dose cytarabine plus fludarabine (FLUGA). * Green: predictors of favorable response. Red: predictors of adverse outcome. HMR: high-molecular-risk pattern defined by the presence of *NRAS* and/or *TP53* mutations. Score predicting an AZA response defined by presence of *DNMT3A*, *TET2*, *EZH2*, or *U2AF1* mutations * subtypes which benefit from the AZA arm vs. LDAC+ fludarabine based on biological and genomic characteristics. AML: acute myeloid leukemia.

## Data Availability

All supporting data are included in the manuscript and Appendix A. The data discussed in this publication have been deposited in the NCBI Sequence Read Archive (SRA) and are accessible with the project ID: PRJNA655113. Additional data are available upon reasonable request from the corresponding author.

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
