# Peer review of "The Mutational Landscape of Acute Myeloid Leukaemia Predicts Responses and Outcomes in Elderly Patients from the PETHEMA-FLUGAZA Phase 3 Clinical Trial"

_cancers, 2021, doi:10.3390/cancers13102458_

Round 1

Reviewer 1 Report

The authors examined and described the mutational landscape of a panel of 43 genes in AML of elderly patients treated with two different regimens. Bascially 3 genes were found to have prognostic significance.

Minor Suggestions for Improving the Manuscript:

(1)  Visual Abstract, upper part (page 2):  Maybe it is possible to highlight the genes BCOR, NRAS and TP53, for example with a red arrow, red asterix or red background (or in a different color).

(2)  Introduction, first line and third line (page 3):  The terms "older patients" and "elderly patients" could be somehow defined, maybe at least approximately.

(3)  Introduction, third paragraph (page 3):  „The clinical outlook for the elderly population has improved over the last few years“. Is a reference (or references) with convincing data available that can here be cited?

(4)  Table 1 (page 4): 

-  The range of ages could be indicated (not only the median).

-  In this table and at many places elsewhere in the text and in the tables, percentages should be limited to the full number and should not include decimals (except maybe for percentages below 10 %, but also here 1 decimal would be sufficient).

-  It is also not necessary to indicate age in years with two decimals or follow-up time in months with two decimals (just full months is sufficient).

-  The list of abbreviations should be in alphabetical order (otherwise one has to look through the whole text in search for a particular abbreviation). Same goes for the supplemental tables.

(5)  Methods, 2.2.1. (page 5):  The single nucleotide polymorphism database had been accessioned mostly recently in November 2019. Have the authors since then consulted this database?

(6)  Methods, 2.2.1. (page 5):  I could not find Suppl. Table 7 which is also not listed on page 2 of the manuscript supplements.

(7)  Figure 3:  Figure 3 appears already on pages 6/7 while Figure 1 appears on page 8.

(8)  Results, 3.2. (page 8):  It is easier to read if the genes are listed in alphabetical order as in the Suppl. Figure 5.

(9)  Legend to Figure 4 (page 10):  The words „predicts Scheme 3. 4. TP53 and NRAS …“ make no sense.

(10)  Discussion, 2nd paragraph (page 11):  „enriched“ (not „enhanced“).

(11)  Discussion, last paragraph (page 12):  Maybe the notion „We have defined molecular signatures for identifying patients …“ could be strengthened by a small summary table or a small summary figure as a succinct take-home message (naming the relevant genes).

(12)  Supplements, page 2, last line:  Suppl. Table 7 was not included.

(13)  Suppl. Table 1 (page 7):  It is curious that Table 1 shows the 2 subgroups while Suppl. Table 1 shows the whole group. Would one table for all 3 categories not be more useful?

(14)  Not cited in the manuscript text are:  Suppl. Table 5, Suppl. Table 6, and Suppl. Figure 1B.

(15)  Legend to Suppl. Figure 1 (page 19):  Why are there periods instead of kommas? Genes with significant differences may be marked with an asterix or a colored background for easier recognition.

(16)  Suppl. Figure 4 (in the box) (page 25):  „global“ (not „glogal“).

(17)  Suppl. Figure 6 (page 27):  Write the genes into each sub-figure, maybe in the upper right quadrant, for easier reading.

Author Response

We especially appreciate all the recommendations for changes to the article.

See the answers in the attached document.

Thanks in advances

Reviewer 2 Report

In the manuscript presented here by Ayala et al, the authors describe their studies of outcome prediction based on genetic mutational profiling.They show that treatment with azacytidine is beneficial for patients with low-moderate cytogenetic risk and for patients with NRAS or TP53 mutations. The publication is well elaborated, statistically valid supported and shows highly interesting results.

Minor comments:
The graphical abstract should be revised. It is illegible and incomprehensible in itself

Author Response

Thank you very much for your comments.

The graphical abstract has been changed.

Reviewer 3 Report

It is important to include molecular studies in clinical trials in order to test new biomarkers for prediction and response. Authors analysed bone marrow DNA samples at the time of diagnosis by high-depth next generation sequencing in a phase 3 trial of azacitidine (AZA) versus low dose cytarabine plus fludarabine (FLUGA) of 207 older AML patients (median 75-76 years at diagnosis in both arms in clinical FLUGAZA trial NCT02319135. Mutational profiles of older patients differed considerably from those of young AML patients treated with high-dose cytarabine plus idarubicin (3+7 regimen). Recurrent genetic abnormalities in genes FLT3-ITD, NPM1, and CEBPA were without impact on overal survival (OS) of older AML patients unlike younger AML patients. Only cytogenetic risk had a prognostic impact on OS. I think that manuscript is acceptable for publication. Only abbreviation RFS (relapse-free survival) should be added in Abbreviations.   

Author Response

Thank you for the assessment and the abbreviations were revised.

Round 2

Reviewer 1 Report

no comments to authors